# Proximal Mapping for Deep Regularization

**Mao Li,     Yingyi Ma,     Xinhua Zhang**
Department of Computer Science, University of Illinois at Chicago
Chicago, IL 60607
{mli206, yma36, zhangx}@uic.edu

## Abstract

Underpinning the success of deep learning is effective regularizations that allow a variety of priors in data to be modeled. For example, robustness to adversarial perturbations, and correlations between multiple modalities. However, most regularizers are specified in terms of hidden layer outputs, which are not themselves optimization variables. In contrast to prevalent methods that optimize them indirectly through model weights, we propose inserting proximal mapping as a new layer to the deep network, which directly and explicitly produces well regularized hidden layer outputs. The resulting technique is shown well connected to kernel warping and dropout, and novel algorithms were developed for robust temporal learning and multiview modeling, both outperforming state-of-the-art methods.

## 1   Introduction

The success of deep learning relies on massive neural networks that often considerably out-scale the training dataset, defying the conventional learning theory [1, 2]. Regularization has been shown essential and a variety of forms are available. For example, invariances to transformations such as rotation [3] have been extended beyond group-based diffeomorphisms to indecipherable transformations that are only exemplified by pairs of views [4], e.g., sentences uttered by the same person. Prior regularities are also commonly available **a)** *within* layers of neural networks, such as sparsity [5], spatial invariance in convolutional nets, structured gradient that accounts for data covariance [6]; **b)** *between* layers of representation, such as stability under dropout and adversarial perturbations of preceding layers [7], contractivity between layers [8], and correlations in hidden layers among multiple views [9, 10]; and **c)** at batch level, e.g., disentangled representation and multiple modalities.

The most prevalent approach to incorporating priors is regularization, which leads to the standard regularized risk minimization (RRM) for a given dataset $\mathcal{D}$, empirical distribution $\tilde{p}$, and loss $\ell$:

$$\min_f \ \mathbb{E}_{x\sim\tilde{p}}[\ell(f(x))] + \Gamma(f) + \sum_i \Omega_i(\{h_i(x,f)\}_{x\in\mathcal{D}}). \tag{1}$$

Here $f$ is the predictor (e.g., neural network), and $\Gamma$ is the data-independent regularizer (e.g., $L_2$ norm), and $\Omega_i$ is the data-dependent regularizer on the $i$-th layer output $h_i$ under $f$ (e.g., invariance of $h_i$ with respect to the $i$-th step input $x_i$ in an RNN). Note $\Omega_i$ can involve multiple layers (e.g., contractivity), or be decomposed over training examples. Optimization techniques such as end-to-end training have produced strong performance, along with progresses in the global analysis of the solution [e.g., 11]. However, all these analyses make assumptions on the landscape of the objective function, which, although often satisfied by the empirical risk $\mathbb{E}_{x\sim\tilde{p}}[\ell(f(x))]$, are typically violated or complicated by the addition of data-dependant regularizers $\Omega_i$. The nontrivial contention between accurate prediction and faithful regularization can often confound the optimization of model weights.

A natural question therefore arises: is it possible to further improve the effectiveness of regularization, potentially not only through the development of new solvers and analysis for RRM, but also through novel mechanisms of incorporating regularization? Although the former approach has been studied intensively, we hypothesize and will demonstrate empirically that the latter approach can

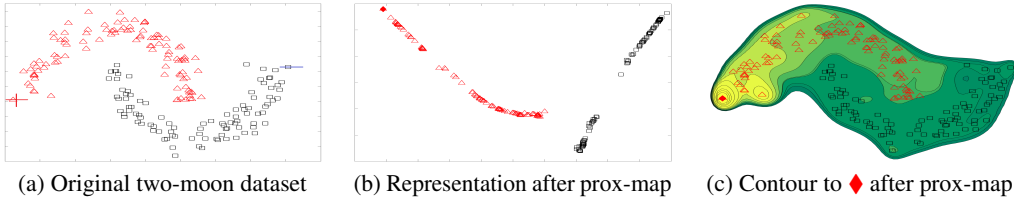

| (a) Original two-moon dataset | (b) Representation after prox-map | (c) Contour to ◆ after prox-map |

Figure 1: (a) The two-moon dataset with only two labeled examples '+' and '−' (left and right), but abundant unlabeled examples that reveal the inherent structure; (b) Representation inferred from top-2 kernel PCA based on the proximal mapping with gradient flatness and Gaussian kernel (see §3); (c) contour of distance to the leftmost point ◆, based on the result of proximal mapping.

be surprisingly effective. Our key intuition is, now that $\Omega_i$ is specified in terms of the hidden layer output $h_i$ (which is determined by $f$), can we directly optimize $h_i$ as opposed to indirectly through $f$? Treating $h_i$ as ground variables and optimizing them jointly with model weights has been used by [12]. However, their motivation is on accelerating the optimization rather than improving the model.

It turns out that this idea can be conveniently implemented by leveraging the tool of proximal mapping (hence the name ProxNet), which has been extensively used in optimization to enforce structured solutions such as sparsity [13]. Given a closed convex set $C \subseteq \mathbb{R}^n$ and a convex function $R : \mathbb{R}^n \to \mathbb{R}$ which favor certain desirable prior (e.g., $\ell_1$ norm), the proximal mapping $\mathsf{P}_R : \mathbb{R}^n \to \mathbb{R}^n$ is defined as

$$\mathsf{P}_R(x) := \arg\min_{z \in C}\{R(z) + \tfrac{\lambda}{2}\|z - x\|^2\}, \quad \text{where the norm is } L_2. \tag{2}$$

In essence, $R$ and $C$ encourage the mapping to respect the prior encoded by $R$, while remaining in the vicinity of $x$. For example, Figure 1a shows the two-moon dataset with only two labeled examples and many unlabeled ones. Figures 1b and 1c show the resulting representation and warped distance where $R$ accounts for the underlying manifold, making the classification trivial (§3).

In a deep network, the proximal mapping can be inserted after any layer to turn $h_i$ into $\mathsf{P}_{\Omega_i}(h_i)$, and backpropagate through it. **Why** does this yield a more effective implementation of regularization? First of all, it provides the modularity of decoupling regularization from supervised learning — the regularization is encapsulated *within* the proximal layer that is *free of weights*, and the resulting $\mathsf{P}_{\Omega_i}(h_i)$ is directly enforced to comply with the prior rather than indirectly through the optimization of weights in $f$. This frees weight optimization from simultaneously catering to unsupervised structures and supervised performance metrics, which plagues the conventional RRM. Such an advantage will be confirmed in our experiments of end-to-end training that are highly efficient (§5.1).

Secondly, proximal mapping can be interpreted as an intermediate step of denoising, where $\mathsf{P}_{\Omega_i}(h_i)$ is a *cleaned* version of $h_i$ that conforms to the prior. This ensures that the downstream layers are presented with well regularized inputs, which will presumably facilitate their own learning. By gradually increasing $\lambda$, such a manual morphing can be annealed, allowing the upstream layers (e.g., feature extractors) to approach weight values that by themselves produce well-regularized $h_i$. ProxNet is also readily connected with meta-learning (§B) because of the bi-level optimization setup, where the proximal layer plays a similar role to base-learners.

Finally, $\mathsf{P}_{\Omega_i}(h_i)$ can be carried out on a mini-batch $\mathcal{B}$, where $R$ is defined on a set $\{h_i(x)\}_{x \in \mathcal{B}}$. It also extends flexibly to regularizers that account for multiple layers, e.g., invariance of $h_i$ to $h_{i-1}$.

This paper will first review the existing building blocks of deep networks through the lens of proximal mapping (§2), and then unravel its non-trivial connections with regularization when the latter is quadratic (e.g., manifold smoothness) or non-quadratic (e.g., dropout). Afterwards, two novel ProxNets will be introduced that achieve robust recurrent modeling (§4) and multiview learning (§5). Extensive experiments show that ProxNet outperforms state-of-the-art prediction models (§6).

**Related Work**   ProxNet instantiates the differentiable optimization framework laid by OptNet [14, 15] along with [16–25], which provides recipes for differentiating through an optimization layer. In contrast, our focus is not on optimization, but on using ProxNet to model the priors in the data, which typically involves an (inner) unsupervised learning task such as CCA. More detailed discussions on the relationship between ProxNet and OptNet or related works are available in Appendix A.

Another proximal-like operator was found in "sparsemap" operations [26–28]. However, they target a different application of incorporating structured sparsity in attention weights for a *single* instance, rather than at a mini-batch level where ProxNet is applied for multiview learning.

## 2  Proximal Mapping as a Primitive Construct in Deep Networks

Proximal mapping is highly general, encompassing most primitive operations in deep learning [13, 29]. For example, any activation function $\sigma$ with $\sigma'(x) \in (0, 1]$ (e.g., sigmoid) is indeed a proximal map with $C = \mathbb{R}^n$ and $R(x) = \int \sigma^{-1}(x) \, \mathrm{d}x - \frac{1}{2}x^2$, which is convex. The ReLU and hard tanh activations can be recovered by $R = 0$, with $C = [0, \infty)$ and $C = [-1, 1]$, respectively. Soft-max transfer $\mathbb{R}^n \ni x \mapsto (e^{x_1}, \dots, e^{x_n})^\top / \sum_i e^{x_i}$ corresponds to $C = \{x \in \mathbb{R}^n_+ : \mathbf{1}^\top x = 1\}$ and $R(x) = \sum_i x_i \log x_i - \frac{1}{2}x_i^2$, which are convex. Batch normalization maps $x \in \mathbb{R}^n$ to $(x - \mu \mathbf{1})/\sigma$, where $\mathbf{1}$ is a vector of all ones, and $\mu$ and $\sigma$ are the mean and standard deviation of the elements in $x$, respectively. This mapping can be recovered by $R = 0$ and $C = \{x : \|x\| = \sqrt{n}, \mathbf{1}^\top x = 0\}$. Although $C$ is not convex, this $\mathsf{P}_R(x)$ must be a singleton for $x \neq 0$. In general, $R$ and $C$ can be nonconvex making $\mathsf{P}_R(z)$ set-valued, and we only need differentiation at one element [30–32].

**Kernelization.**  Proximal mapping can be trivially extended to reproducing kernel Hilbert spaces (RKHSs), allowing non-vectorial data to be encoded [33] and invariances to be hard wired [34, 35]. Assume an RKHS $\mathcal{H}$ employs a kernel $k : \mathcal{X} \times \mathcal{X} \to \mathbb{R}$ with an inner product $\langle \cdot, \cdot \rangle_{\mathcal{H}}$. Given a convex functional $R : \mathcal{H} \to \mathbb{R}$, a proximal map $\mathsf{P}_R : \mathcal{H} \to \mathcal{H}$ can be defined in exactly the same form as (2), with $L_2$ norm replaced by RKHS norm.

## 3  Connecting Proximal Mapping to RRM on Shallow Models

We first illustrate the connection between RRM and proximal mapping. To focus on the core idea, we use shallow models with no hidden layer. Letting $k_x := k(x, \cdot)$ be the kernel representer of $x$ and $R$ be the regularizer encoding preference on $f$, we can write the two formulations as follows:

$$\text{P1:} \ \min_{f \in \mathcal{H}} \mathbb{E}_{x \sim \tilde{p}}[\ell(\langle f, k_x \rangle_{\mathcal{H}})] + R(f) \quad \text{v.s.} \quad \text{P2:} \ \min_{h \in \mathcal{H}} \mathbb{E}_{x \sim \tilde{p}}[\ell(\langle h, c_x \rangle_{\mathcal{H}})] + \lambda^2 \|h\|_{\mathcal{H}}^2 , \quad (3)$$

$$\text{where} \quad c_x := \mathsf{P}_R(k_x) = \arg\min_{g \in \mathcal{H}} \left\{ \frac{\lambda}{2} \|g - k_x\|_{\mathcal{H}}^2 + R(g) \right\}. \quad (4)$$

**i) $R(f)$ is a positive semi-definite (PSD) quadratic.**  Examples of this simplest case include graph Lapalacian $R_l(f) := \sum_{ij} w_{ij}(f(x_i) - f(x_j))^2$ and gradient penalty $R_g(f) := \sum_i \|\nabla f(x_i)\|^2$. They both enforce smoothness on a data manifold. Since the gradient operator $\nabla R : f \mapsto \nabla R(f)$ is linear, we denote its eigenvalues and eigenfunctions as $\{\mu_i, \phi_i\}$. Further, taking derivative of $g$ in (4), we derive a closed-form for the proximal map as $c_x = \lambda(\lambda I + \nabla R)^{-1} k_x$, where $I$ is the identity operator. The contour in Figure 1c was plotted exactly by using the pairwise distance $\|c_x - c_{x'}\|_{\mathcal{H}}$, based on which the new data representation in Figure 1b was extracted using the top-2 principal components.

To connect P1 and P2, let $h = \lambda^{-1}(\lambda I + \nabla R)f$. Then $\langle f, k_x \rangle_{\mathcal{H}} = \langle h, c_x \rangle_{\mathcal{H}}$ (i.e., same prediction) and

$$R(f) = \frac{1}{2} \sum_i \mu_i \langle f, \phi_i \rangle_{\mathcal{H}}^2, \qquad \text{and} \qquad \lambda^2 \|h\|_{\mathcal{H}}^2 = \sum_i (\lambda + \mu_i)^2 \langle f, \phi_i \rangle_{\mathcal{H}}^2. \quad (5)$$

This reveals that P1 and P2 are connected through a *monotonic* spectral transformation. When $\lambda$ is small, it simply squares the eigenvalues, which leads to little difference in learning as we observed in experiment. Moreover, there is a similar connection between $c_x$ and the kernel representer of a new RKHS, which warps the original RKHS norm into $\|f\|_{\mathcal{H}}^2 + R(f)$ [36]. See details in Appendix C.

**ii) General $R$.**  When $R$ is not quadratic, the linear relationship between $c_x$ and $k_x$ no longer exists. However, some relaxed connection between P1 and P2 is still available, and we will demonstrate it on dropout training. As discovered by [37, 38], dropout on input features in a single-layer network leads to an adaptive regularizer on a linear discriminant $x \mapsto \beta^\top x$ (derivation is in Appendix D):

$$R_{\tilde{p}}(\beta) = \sum_i \mathbb{E}_{x \sim \tilde{p}}[p_x(1 - p_x)x_i^2] \cdot \beta_i^2, \quad \text{where} \quad p_x := \sigma(x^\top \beta) := (1 + \exp(-x^\top \beta))^{-1}. \quad (6)$$

Here $R_{\tilde{p}}$ penalizes $\beta_i$ more mildly if $x_i$ is generally small. This allows rare but discriminative features to receive higher weights, which is useful in text data. Now to connect P1 and P2, we simplify the

computation by using the proximal map of $R_{\delta_x}$ instead of $R_{\tilde{p}}$, where $\delta_x$ is the Dirac distribution at $x$:

$$c_x := \mathsf{P}_{R_{\delta_x}}(x) = \arg\min_c \left\{ \tfrac{1}{2} \sum_i p_x(1-p_x) x_i^2 c_i^2 + \tfrac{\lambda}{2} \|c - x\|^2 \right\}, \text{ where } p_x = \sigma(x^\top c). \quad (7)$$

Since $p_x$ depends only on $x^\top c$, we first fix $x^\top c$ to $s$, hence $p_x(1-p_x) = \alpha_s := \frac{2}{2+e^s+e^{-s}}$. Enforcing $x^\top c = s$ by a Lagrange multiplier $\mu$, $c_i$'s are decoupled, allowing them to be optimized analytically:

$$(c_x)_i = (\lambda + \mu) x_i (\lambda + \alpha_s x_i^2)^{-1}, \qquad \text{where} \quad \mu \text{ is such that } x^\top c_x = s. \quad (8)$$

Finally (7) can be optimized through a 1-D line search on $s$. Letting $h_i = \beta_i \frac{\lambda + \alpha_s x_i^2}{\lambda + \mu}$, we have $\beta^\top x = h^\top c_x$ (same predictions) and $\|h\|^2 = (\lambda+\mu)^{-2} \sum_i (\lambda + p_x(1-p_x) x_i^2)^2 \beta_i^2$, which resembles $R_{\tilde{p}}$ in (6) especially when $\lambda$ is small. However, since $\frac{h_i}{\beta_i}$ depends on $x$, this reformulation meets with difficulty when extended to the whole dataset $\tilde{p}$. We emphasize that our aim here is to shed light on the connection between regularization and proximal mapping; we do *not* intend to establish their exact equivalence. Simulations in Appendix D show that P1 and P2 deliver very similar predictions.

**Application to multiple layers.** It is straightforward to apply proximal mapping to any hidden layer of interest and for multiple times. A similar warping trick was introduced in [36] to invariantize convolutional kernel descriptors [34, 39]. However it was restricted to linear invariances. Proximal mapping, instead, lifts this restriction by accommodating nonlinear invariances such as total variation.

## 4 Proximal Mapping for Robust Learning in Recurrent Neural Nets

Our first novel instance of ProxNet tries to invariantize LSTM to perturbations on inputs $x_t$. Virtual adversarial training has been proposed in this context as an unsupervised regularizer [40], where the underlying prior postulates that such robustness can benefit the prediction accuracy. The resilience under real attack, however, is *not* the main concern in [40]. We will demonstrate empirically that this prior can be more effectively implemented by ProxNet, leading to improved prediction performance.

The dynamics of hidden states $c_t$ in an LSTM can be represented by $c_t = f(c_{t-1}, h_{t-1}, x_t)$, with outputs $h_t$ updated by $h_t = g(c_{t-1}, h_{t-1}, x_t)$. We aim to encourage that the hidden state $c_t$ stays invariant, when each $x_t$ is perturbed by $\delta_t$ whose norm is bounded by $\delta$. To this end, we introduce an intermediate step $s_t = s_t(c_{t-1}, h_{t-1}, x_t)$ that computes the original hidden state, and then apply proximal mapping so that the next state $c_t$ remains close to $s_t$, while also moving towards the *null space* of the variation of $s_t$ under the perturbations on $x_t$. Formally, using first-order approximation,

$$c_t := \arg\min_c \tfrac{\lambda}{2} \|c - s_t\|^2 + \tfrac{1}{2} \max_{\|\delta_t\| \le \delta} \langle c, s_t(c_{t-1}, h_{t-1}, x_t) - s_t(c_{t-1}, h_{t-1}, x_t + \delta_t) \rangle^2$$

$$\approx \arg\min_c \lambda \|c - s_t\|^2 + \max_{\|\delta_t\| \le \delta} \langle c, \tfrac{\partial}{\partial x_t} s_t(c_{t-1}, h_{t-1}, x_t) \delta_t \rangle^2$$

$$= \arg\min_c \lambda \|c - s_t\|^2 + \delta^2 \|c^\top G_t\|_*^2, \quad \text{where} \quad G_t := \tfrac{\partial}{\partial x_t} s_t(c_{t-1}, h_{t-1}, x_t)$$

and $\|\cdot\|_*$ is the dual norm. The diagram is shown in Figure 2. Using the $L_2$ norm, a closed-form solution for $c_t$ is $(I + \lambda^{-1}\delta^2 G_t G_t^\top)^{-1} s_t$, and BP can be reduced to second-order derivatives (§F). A key advantage of this framework is the generality and ease in inserting proximal layers into the framework — simply invoke the second-order derivatives of the underlying (gated) units as a black box, be it LSTM or GRU. We will refer to this model as ProxLSTM. To summarize, we achieved robustness not because of using the recurrent structure itself, but by properly invariantizing each step via embedding a proximal mapping, which is innately synergistic with the recurrent structure. So this technique can be generically deployed in neural networks.

## 5 ProxNet for Multiview Learning

While proximal mapping is applied on each individual data point in ProxLSTM, it can indeed be applied in mini-batches, and we next demonstrate its application in multiview learning with sequential structures. Here each instance exhibits a pair of views: $\{(x_i, y_i)\}_{i=1}^n$, and is associated with a label $c_i$. In the deep canonical correlation analysis model [DCCA, 10], the $x$-view is passed through a multi-layer neural network or kernel machine, leading to a hidden representation $f(x_i)$. Similarly the $y$-view is transformed into $g(y_i)$. CCA aims to maximize the correlation of these two views

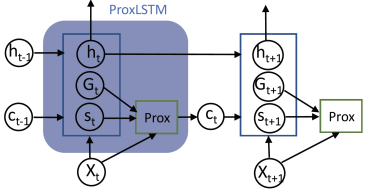

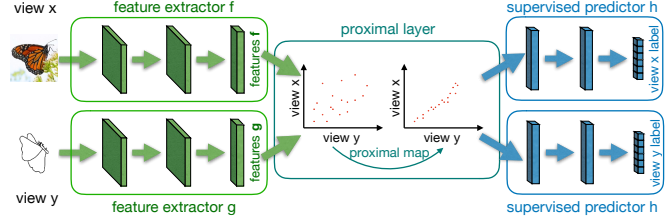

Figure 2: A proximal LSTM layer    Figure 3: ProxNet for multiview learning with proximal CCA

after projecting into a common $k$-dimensional subspace, through $\{u_i\}_{i=1}^k$ and $\{v_i\}_{i=1}^k$ respectively. Denoting $X = (f(x_1), \ldots, f(x_n))H$ and $Y = (g(y_1), \ldots, g(y_n))H$ where $H = I - \frac{1}{n}\mathbf{1}\mathbf{1}^\top$ is the centering matrix, CCA finds $U = (u_1, \ldots, u_k)$ and $V = (v_1, \ldots, v_k)$ that maximize the correlation:

$$\min_{U,V} -\operatorname{tr}(U^\top X Y^\top V), \tag{9}$$

$$\text{s.t. } U^\top X X^\top U = I,\ V^\top Y Y^\top V = I,\ u_i^\top X Y^\top v_j = 0,\ \forall i \neq j. \tag{10}$$

Denote the optimal objective value as $L(X, Y)$. DCCA directly optimizes it with respect to the parameters in $f$ and $g$, while DCCA autoencoder [DCCAE, 9] further reconstructs the input. They both use the result to initialize a finer tuning of $f$ and $g$, in conjunction with subsequent layers $h$ for a supervised target $c_i$. We aim to improve this two-stage process with an end-to-end approach based on proximal mapping, which can be written as $\min_{f,g,h} \sum_i \ell(h(p_i, q_i), c_i)$ where $\{(p_i, q_i)\}_{i=1}^n$ is from

$$\mathsf{P}_L(X, Y) = \arg\min_{P,Q}\ \frac{\lambda}{2n}\|P - X\|_F^2 + \frac{\lambda}{2n}\|Q - Y\|_F^2 + L(P, Q). \tag{11}$$

Here $\|\cdot\|_F$ stands for the Frobenius norm, $P = (p_1, \ldots, p_n)$, and $Q = (q_1, \ldots, q_n)$. Clearly, (11) has applied proximal mapping to a *mini-batch*, and we will show how to save computational cost, especially at test time. The entire framework is illustrated in Figure 3.

## 5.1   Backpropagation and computational cost

Although efficient closed-form solution is available for the CCA objective in (9), none exists for the proximal mapping in (11). However, it is natural to take advantage of this closed-form solution. In particular, assuming $f(x_i)$ and $g(y_i)$ have the same dimensionality, [10] showed that $L(X, Y) = -\sum_{i=1}^k \sigma_i(T)$, where $\sigma_i$ is the $i$-th largest singular value, and

$$T(X, Y) = (XX^\top + \epsilon I)^{-1/2}(XY^\top)(YY^\top + \epsilon I)^{-1/2}.$$

Here $\epsilon > 0$ is a small stabilizing constant. Then (11) can be solved by gradient descent or L-BFGS. The gradient of $\sum_{i=1}^k \sigma_i(T(P, Q))$ is available from [10], which relies on SVD. Although SVD appears expensive, fortunately, the cost of computing $T$ and SVD is low in practice because i) the dimensions of $f$ and $g$ are low in practice (10 in our experiment and DCCA), and ii) the mini-batch size does not need to be large. In our experiment, increasing mini-batch size beyond 100 did not significantly improve the performance. Extension to more than two views is relegated to Appendix E.

Backpropagation through the proximal mapping in (11) requires that given $\frac{\partial J}{\partial P}$ and $\frac{\partial J}{\partial Q}$ where $J$ is the ultimate objective value, compute $\frac{\partial J}{\partial X}$ and $\frac{\partial J}{\partial Y}$. The most general solution has been provided by OptNet [14, 15], but the structure of our problem admits a simpler solution from [16].

$$\left(\tfrac{\partial J}{\partial X}, \tfrac{\partial J}{\partial Y}\right) \approx \tfrac{1}{\epsilon}\left(\mathsf{P}_L(X + \epsilon\tfrac{\partial J}{\partial P}, Y + \epsilon\tfrac{\partial J}{\partial Q}) - \mathsf{P}_L(X, Y)\right), \qquad 0 < \epsilon \ll 1. \tag{12}$$

[16] provided several heuristics for setting $\epsilon$. We set $\epsilon = \delta(1 + \|(X, Y)\|_\infty)\|(\tfrac{\partial L}{\partial P}, \tfrac{\partial L}{\partial Q})\|_\infty^{-1}$ in our experiments so as to be adaptive to the magnitude of the gradient, and $\delta$ is a small constant whose value was chosen by checking a few gradients at the beginning of training. To estimate the "true" gradient needed for the check, we used a small $\epsilon$ and solved the proximal mapping in (11) to high accuracy. With this heuristic, the approximate gradient did not cause instability in training. It is also noteworthy that the stochastic gradient computed from a mini-batch introduces noise in the first place.

To reduce the test time complexity for ProxNet, we draw a key insight that if the feature extractor preceding the proximal mapping is well trained so that the latent representation of the two views is highly correlated, then the proximal layer may improve performance only marginally.

Figure 4: ProxNet for multiview sequential data

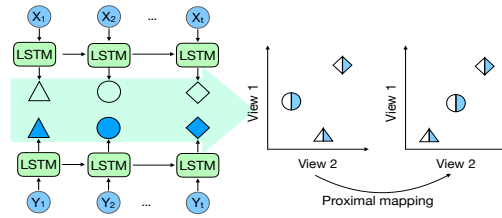

Table 1: Average test error (%) on Sketchy

| #class | 20 | 50 | 100 | 125 |
|---|---|---|---|---|
| Vanilla | $18.7 \pm 1.1$ | $24.8 \pm 0.9$ | $30.9 \pm 0.5$ | $31.8 \pm 0.5$ |
| DCCA | $16.9 \pm 0.5$ | $22.2 \pm 0.4$ | $28.7 \pm 0.4$ | $29.8 \pm 0.4$ |
| DCCAE | $16.6 \pm 0.3$ | $22.1 \pm 0.3$ | $29.2 \pm 0.5$ | $30.4 \pm 0.6$ |
| RRM | $15.2 \pm 0.6$ | $20.1 \pm 0.4$ | $26.8 \pm 0.5$ | $28.1 \pm 0.4$ |
| $\text{RRM}^{anl}$ | $17.4 \pm 0.8$ | $22.5 \pm 0.5$ | $24.3 \pm 0.9$ | $26.1 \pm 0.7$ |
| ProxNet | $\mathbf{13.7} \pm 0.3$ | $\mathbf{17.9} \pm 0.5$ | $\mathbf{20.2} \pm 0.3$ | $\mathbf{22.0} \pm 0.4$ |

Therefore, we can take advantage of proximal mapping during training, while gradually fade it out at the fine tuning stage. Towards this end, the weight $\lambda$ that controls the trade off between correlation and displacement can be increased as training proceeds. More specifically, we set in experiment $\lambda_t = (1 + kt)\alpha_0$ at epoch $t$, where $\alpha_0$ and $k$ are hyperparameters. As a result, test time predictions can be made very efficiently by dispensing with proximal mapping or mini-batch.

## 6   Experimental Results

We evaluated the empirical performance of ProxNet for multiview learning on supervised learning (two tasks) and unsupervised learning (crosslingual word embedding). ProxLSTM was evaluated on sequence classification. We used the Ray Tune library to select the hyper-parameters for all baseline methods [41]. Details on data preprocessing, experiment setting, optimization, and additional results are given in Appendix G. Here we highlight the major results and experiment setup.

All code and data are available at `https://github.com/learndeep2019/ProxNet`.

**Baselines.** For the three multiview tasks, we will demonstrate the effectiveness of ProxNet by comparing with state-of-the-art methods including **DCCA** and **DCCAE**. Neither DCCA nor DCCAE is end-to-end training, and a classifier was trained on their hidden code. As a basic competitor, we also considered a **Vanilla** method, which trained a network for each view independently.

Our key competitor is RRM, which motivated ProxNet in the introduction section. Specifically, it moves the regularizer $L(X, Y)$ defined in (9) from inside the proximal mapping to the overall objective as in (1), promoting the correlation between the two views' hidden representation through the network weights. At test time, ProxNet, RRM, and Vanilla all made predictions by averaging the logits from both views. This consistently outperformed concatenating the logits of the two views.

### 6.1   Multiview supervised learning 1: image recognition with sketch and photo

**Dataset.** We first evaluated ProxNet on a large-scale sketch-photo paired database "Sketchy" [42]. It consists of 12,500 photos and 75,471 hand-drawn sketches of objects from 125 classes. Each instance is a pair of sketch and photo representing the same natural image, both in color and sized $256 \times 256$. To demonstrate the robustness of our method, we varied the number of classes over $\{20, 50, 100, 125\}$ by sampling a subset from the original dataset. For each class, there are 100 sketch-photo pairs. We randomly sampled 80 pairs from each class to form the training set, and then used the remaining 20 pairs for testing.

**Implementation details.** Unless otherwise specified, our implementations were based on PyTorch and all training was conducted on a NVIDIA GeForce 2080 Ti GPU. All methods were trained using ResNet-18 as the feature extractor. In ProxNet, the proximal layer has input and output dimension $d = 20$, followed by three fully-connected layers of 512 hidden units with sigmoid activations. The final output layer has multiple softmax units, each corresponding to an output class. ProxNet was trained by Adam with a weight decay of 0.0001 and a learning rate of 0.001, with the latter divided by 10 after 200 epochs. The mini-batch size was 100, which, in conjunction with the low dimensionality of proximal layer ($d = 20$), allows the SVD to be solved instantaneously. At training time, we employed an adaptive trade-off parameter $\lambda$, which is defined in (11). We set the hyper-parameters $k = 0.5$ and $\alpha_0 = 0.1$. All experiments were run five times to produce mean and standard deviation.

Table 2: Mean and standard deviation of PERs on the XRMB dataset with different noise levels

| | noise level = 0% | | noise level = 20% | | noise level = 50% | | noise level = 80% | |
|---|---|---|---|---|---|---|---|---|
| | acoustic | logit-avg | acoustic | logit-avg | acoustic | logit-avg | acoustic | logit-avg |
| Vanilla | $17.9 \pm 1.0$ | $17.1 \pm 0.6$ | $19.3 \pm 0.8$ | $19.1 \pm 1.2$ | $27.7 \pm 1.1$ | $21.4 \pm 0.8$ | $45.1 \pm 1.0$ | $24.4 \pm 1.0$ |
| DCCA | $17.3 \pm 0.3$ | $16.3 \pm 0.5$ | $18.8 \pm 0.3$ | $16.3 \pm 0.6$ | $26.0 \pm 0.3$ | $23.6 \pm 0.5$ | $45.1 \pm 0.9$ | $34.9 \pm 0.9$ |
| DCCAE | $15.5 \pm 0.2$ | $15.3 \pm 0.4$ | $16.7 \pm 0.3$ | $15.9 \pm 0.4$ | $23.6 \pm 0.3$ | $21.8 \pm 0.7$ | $43.9 \pm 0.7$ | $34.8 \pm 0.7$ |
| RRM | $16.1 \pm 0.5$ | $15.0 \pm 0.3$ | $16.6 \pm 0.7$ | $16.9 \pm 0.5$ | $\mathbf{22.3} \pm 0.8$ | $21.6 \pm 0.6$ | $40.7 \pm 0.7$ | $23.9 \pm 0.3$ |
| ProxNet | $\mathbf{12.9} \pm 0.4$ | $\mathbf{10.5} \pm 0.4$ | $\mathbf{15.3} \pm 0.4$ | $\mathbf{11.2} \pm 0.3$ | $21.6 \pm 0.5$ | $\mathbf{16.6} \pm 0.3$ | $\mathbf{39.3} \pm 0.3$ | $\mathbf{20.1} \pm 0.5$ |

**Results.** As shown in Table 1, ProxNet delivers significantly lower test error than all other baselines. Interestingly, the improvement becomes more significant with the increasing number of classes. Vanilla performs the worst, and RRM outperforms DCCA and DCCAE thanks to end-to-end training.

Since we annealed the weight $\lambda$ in the proximal mapping (11) (Section 5.1), it is natural to investigate whether an annealed regularization weight can improve the performance of RRM. Therefore, we conducted additional experiments for it, and the resulting average test error is also presented in Table 1 as $RRM^{anl}$. Clearly, the influence on the vanilla RRM is mixed, but it remains inferior to ProxNet.

## 6.2 Multiview supervised learning 2: audio-visual speech recognition

Our second task aims to learn features and classifiers for speaker-independent phonetic recognition.

**Dataset.** We used the Wisconsin X-ray Micro-Beam Database (XRMB) corpus which consists of simultaneously recorded *speech* and *articulatory* measurements from 47 American English speakers and 2357 utterances [43]. The first view is acoustic features comprising 39D mel frequency cepstral coefficients (MFCCs) and their first and second derivatives, and the second view is articulatory features made up of 16D horizontal/vertical displacement of 8 pellets attached to several parts of the vocal tract. Also available is the phonetic labels for classification. To simulate the real-life scenarios and to improve the model's robustness to noise, we corrupted the acoustic features of a given speaker by mixing with $\{0.2, 0.5, 0.8\}$ level of another randomly picked speaker's acoustic features. The whole dataset was partitioned into 35 speakers for training and 12 speakers for testing.

**Implementation details.** To incorporate context information, [9] concatenated the inputs over a window sized $W$ centered at each frame, giving $39 \times W$ and $16 \times W$ feature dimensions for each of the two views respectively. Although this delicately constructed input freed the encoder/feature extractor from considering the time dependency within frames, we prefer a more refined modeling of the sequential structure, and therefore adopted, for all methods under comparison, a 2-layer LSTM with hidden layers of 256 units, followed by a fully-connected layer which projects the outputs of LSTM to a $K$-dimensional subspace, serving as the feature extractor.

The supervised predictor is a fully-connected network with an output layer of 41 softmax units. We used the Connectionist Temporal Classification loss [CTC, 44], which adopts greedy search as the phone recognizer. The dimension of subspace was tuned in $\{10, 20, 30, 50\}$, and the sequence length was tuned in $\{250, 500, 1000\}$ for all algorithms. The mini-batch size was set to 32. Although the proximal mapping here solves a larger problem than that in §6.1, we observed that a higher value of $\lambda$ was sufficient to enforce a high correlation on this dataset, hence keeping the optimization efficient. In practice, we set $k = 1$ and $\alpha_0 = 0.5$.

In order to compare the effectiveness of different algorithms in information transfer without being confounded by logit averaging (logit-avg) which can achieve a similar effect, we studied another mode called "acoustic". Here all algorithms predict on test data by only using the output layer of the acoustic view, and at training time a loss is applied to each view based on the ground truth label.

**Results.** Table 2 presents the Phone Error Rates (PERs) of all methods. Clearly, ProxNet achieves the lowest PER among all algorithms at all levels of noise. The margin over the runner-up (RRM) is the largest when there is no noise. As expected, "logit-avg" almost always outperforms "acoustic", because the articulatory features are clean, supplying reliable predictions. Focusing on the "acous-

Table 3: Spearman's correlation for word similarity. Following [45], for each algorithm, the model with the highest Spearman's correlation on the 649 tuning bigram pairs was selected.

|  | WS-353 | | WS-SIM | | WS-REL | | SimLex999 | |
| --- | --- | --- | --- | --- | --- | --- | --- | --- |
|  | EN | DE | EN | DE | EN | DE | EN | DE |
| Baseline | 73.4 | 52.7 | 77.8 | 63.3 | 67.7 | 44.2 | 37.2 | 29.1 |
| LinCCA | 73.8 | 68.5 | 76.1 | 73.0 | 67.0 | 62.9 | 37.8 | 43.3 |
| DCCA | 73.9 | 69.1 | **78.7** | 74.1 | 66.6 | 64.7 | 38.78 | 43.29 |
| DCCAE / RRM | 72.4 | **69.7** | 75.7 | 74.7 | 65.9 | 64.2 | 36.7 | 41.8 |
| ProxNet | **75.4** | 69.2 | 78.3 | **75.4** | **71.0** | **66.8** | **40.0** | **44.2** |
| CL-DEPEMB | - | - | - | - | - | - | 35.6 | 30.6 |

tic" columns, Vanilla cannot leverage articulatory features, while other methods can achieve it by promoting correlations in the hidden space. ProxNet appears most effective in this respect.

## 6.3 Multiview unsupervised learning: crosslingual word embedding

We next seek to learn word representations that reflect word similarity, and the multiview approach trains on (English, German) word pairs, hoping that information is transferred in the latent subspace.

**Dataset.** We obtained 36K pairs of English-German word as training examples from the parallel news commentary corpora [WMT 2012-2018, 46], using the word alignment method from [47] and [48]. Based on the corpora we also built a bilingual dictionary, where each English word is matched with the (unique) German word that has been most frequently aligned to it. The raw word embedding ($x_i$ and $y_i$) used the pretrained monolingual 300-dimensional word vectors from fastText [49, 50].

The evaluation was conducted on two commonly used datasets [51, 52]: **a)** multilingual WS353 contains 353 pairs of English words, and their translations to German, Italian and Russian, that have been assigned similarity ratings by humans. It was further split into multilingual WS-SIM and multilingual WS-REL which measure similarity and relatedness between word pairs, respectively; **b)** multilingual SimLex999 consists of 999 English word pairs and their translations.

**Algorithms.** All methods used multilayer perceptrons with ReLU activation. ProxNet used the input reconstruction error as the ultimate objective. As a result, *DCCAE is exactly the RRM variant*. A validation set was employed to select the hidden dimension $h$ for $f$ and $g$ from $\{0.1, 0.3, 0.5, 0.7, 0.9\} \times 300$, the regularization parameter $\lambda$, and the depth and layer width from 1 to 4 and $\{256, 512, 1024, 2048\}$, respectively. We searched the mini-batch size in $\{100, 200, 300, 400\}$. We also compared with linear CCA [53]. At test time, the (English, German) word pairs from the test set were fed to the four multiview based models, extracting the English and German word representations. Then the cosine similarity can be computed between all pairs of monolingual words in the test set (English and German), and we reported in Table 3 the Spearman's correlation between the model's ranking and human's ranking.

**Results.** Clearly, ProxNet always achieves the highest or close to highest Spearman's correlation on all test sets and for both English (EN) and German (DE). We also included a baseline which only uses the monolingual word vectors. CL-DEPEMB is from [54], and the paper only provided the results for SimLex999 with no code made available. It can be observed from Table 3 that multiview based methods achieved more significant improvement over the baseline on German data than on English data. This is not surprising, because the presence of multiple views offers an opportunity to transfer useful information from other views/languages. Since the performance on English is generally better than that of German, more improvement is expected on German.

## 6.4 Robust training for recurrent networks

We now present the experimental results of robust training for LSTMs as described in Section 4.

Table 4: Test accuracy for sequence classification. "len" stands for the median length of the sequences.

|  | #train | len | LSTM | AdvLSTM | ProxLSTM |
|---|---|---|---|---|---|
| JV | 225 | 15 | 94.02 ±0.72 | 94.96 ±0.44 | **95.52** ±0.63 |
| HAR | 6.1k | 128 | 89.75 ±0.59 | **92.01** ±0.18 | **92.08** ±0.23 |
| AD | 5.5k | 39 | 96.32 ±0.55 | 97.45 ±0.38 | **97.99** ±0.29 |
| IMDB | 25k | 239 | 92.65 ±0.04 | 93.65 ±0.03 | **94.16** ±0.11 |

Figure 5: t-SNE embedding of HAR dataset (best viewed in color)

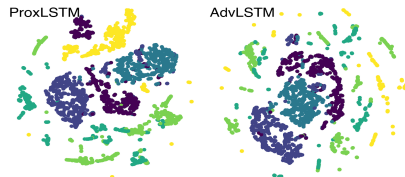

**Datasets.** We tested on four sequence datasets: Japanese vowels [JV, 55] which contains time series data for speaker recognition based on uttered vowels; Human Activity Recognition [HAR, 56] which classifies activity; Arabic Digits [AD, 57] which recognizes digits from speeches; and IMDB [58], a large movie review dataset for sentiment classification. Table 4 presents the training set size and *median* sequence length.

**Algorithms.** We compared ProxLSTM with two baselines: vanilla LSTM and the adversarial training of LSTM [40], which we will refer to as AdvLSTM. For JV, HAR, AD datasets, the base models are preceded by a CNN layer, and succeeded by a fully connected layer. The CNN layer consists of kernels sized 3, 8, 3 and contains 32, 64, 64 filters for the JV, HAR, AD datasets, respectively. LSTM used 64, 128, 64 hidden units for these three datasets, respectively. All these parameters were tuned to optimize the performance of vanilla LSTM, and then shared with ProxLSTM and AdvLSTM for a fair comparison. We first trained the vanilla LSTM to convergence, and used the resulting model to initialize AdvLSTM and ProxLSTM. For IMDB, we first trained AdvLSTM by following the settings in [40], and then used the result to initialize the weights of ProxLSTM. All settings were evaluated 10 times to report mean and standard deviation.

**Results.** From Table 4, it is clear that adversarial training improves test accuracy, and ProxLSTM promotes the performance even more than AdvLSTM. Since the accuracy gap is lowest on the HAR dataset, we also plotted the t-SNE embedding of the features from the *last time step* for HAR. As Figure 5 shows, the representation learned by ProxLSTM is better clustered than that of AdvLSTM, especially the yellow class. This further indicates that ProxLSTM learns better latent representations than AdvLSTM by applying proximal mapping. Plots for other datasets are in §G.4.

## 7 Conclusion

In this paper, we proposed using proximal mapping as a new primitive in deep networks to explicitly encode the prior for end-to-end training. Connection to existing constructs in deep learning are shown. The new model is extended to multiview learning and robust RNNs, and its effectiveness is demonstrated in experiments. It is noteworthy that ProxNet is a means of enforcing deep regularization, while itself does not introduce any new regularizer per se; the regularizer $R$ is to be designed by the practitioners for the specific application, e.g., CCA for multi-view learning. Implementing ProxNet is straightforward as shown in (2) for any generic $R$, though some parameters need to be chosen.

The purpose of the paper is to show that for regularizers defined in terms of hidden layer outputs, it is more effective to regularize **in-place** through a proximal mapping at that layer, compared with adding the regularizer to the overall objective and relying on backpropagation for optimization. By "more effective", we have compared by using the test performance instead of the training objective value, because unlike comparing two different nonconvex solvers, ProxNet results in a different objective than regularized risk minimization.

A rigorous analysis beyond the intuition of in-place regularization will be interesting in the deep context, and we plan to investigate it in the future. We will also apply ProxNet to reinforcement learning with knowledge transfer. Furthermore, it will be very interesting to study the use of multiple proximal mappings at different layers for diverse purposes, e.g., to enforce equivariance in each layer of feature extractor by using the violation as the regularizer $R$, disentanglement at a certain layer, and fairness in prediction.

## Acknowledgements

We thank the reviewers for their constructive comments. This work is supported by Google Cloud and NSF grant RI:1910146. The experiments used the Extreme Science and Engineering Discovery Environment (XSEDE) at the PSC GPU-AI. We gratefully acknowledge the support of NVIDIA Corporation with the donation of the Titan V GPU used for this research.

## Broader Impact

Information can be presented in multiple sensory modalities like vision, sound, and touch. However, many machine/deep learning algorithms are still trained on single-modality data instead of taking full advantage of multiple modalities. Recent works have shown that these applications learning from single-modality data might imperil the model by producing biased and/or even unfair results. For instance, a model trained on image data, which has little or no effect on the acoustic and tactile properties of the imaged scene, might has a different or opposite understand of its semantics which results in a wrong prediction [59]. Our work aims to mitigate this problem by learning representations that capture information shared between two views. Thanks to its flexibility and efficiency of ProxNet, this technique can also be extended to many other applications that may affect our daily life.

**Temporal multiview learning for brain network analysis and mortality forecasting.** In neurological disorder analysis, fMRI and diffusion tensor imaging provide different views of the same brain, and each of them represents a graph of brain regions, evolving over the duration of scanning. This evolution can be modeled as dynamic graphs. Naturally, we can apply the multiview ProxNet to graph convolutional networks to discover salient representations that provide insightful interpretations for medical practitioners. Similar techniques also can be used to understand mortality rates from multiple populations (views) with a time-series structure.

**Adversarial recurrent networks for sentiment analysis on reviews.** In sentiment analysis of reviews, a writer can outwit the auto-rater by changing the style, such as inserting common words and avoiding specific keywords consistently over time. Interestingly, invariance to such perturbations in a sequential model of text can be effectively modeled by inserting a proximal layer at each step to build adversarial ProxNet based on LSTMs.

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
