[Supplementary Material]

# Supplementary Material

## A  Relationship with OptNet and Implicit Differentiation Based Learning

Given a prediction model such as linear model, energy-based model, kernel function, deep neural network, etc, a loss function is needed to measure the quality of its prediction against the given ground truth. Although surrogate losses had been popular in making the loss convex, recently it is often observed that directly comparing the prediction of the model, typically computed through an argmin optimization (or argmax), against the ground truth under the true loss of interest can be much more effective. The error signal is originated from the *last step* through the argmin, and then backpropagated through the model itself for training. For example, Amos et al used it to train input convex neural networks at ICML 2017, [18] used it to train a structured prediction energy network, and [60] used it to train an energy-based model for time-series imputation. Other works include [24, 61], etc. A number of implicit and auto-differentiation algorithms have been proposed for it, e.g., [14, 16, 17, 19, 25].

Other uses of such differentiable optimization have been found in learning attention models [26], meta-learning to differentiate through the base learning algorithm [21, 22], or to train the generator in a generative adversarial model by optimizing out the discriminator [20], or for end-to-end planning and control [23]. In all these cases, differentiable optimization is used as an algorithm to train a *given* component within a multi-component learning paradigm. But each component itself has its own pre-fixed model and parameterization.

To the best of our knowledge, OptNet [14] proposed for the first time using optimization as a *layer* of the deep neural network, hence extending the model itself. However, it focused on efficient algorithms for differentiation[1] and the general framework of optimization layer was demonstrated by using standard operations such as total variation denoising, which bears resemblance to task-driven dictionary learning [62, 63]. It remains unclear how to leverage the general framework of OptNet to flexibly model a broad range of structures, while reusing the existing primitives in deep learning (like our extension of LSTM in Section 4).

This is achieved by ProxNet. Although ProxNet also inserts a new layer, it provides *concrete and novel* ways to model structured priors in data through proximal mapping. [64] used proximal operators for regularizing inverse imaging problems. Most aforementioned works use differentiable optimization as a learning algorithm for a *given* model, while ProxNet uses it as a first-class modeling construct within a deep network. Designing the potential function $f$ in (2) can be highly nontrivial, as we have demonstrated in the examples of dropout, kernel warping, multiview learning, and LSTM.

[65] used proximal mapping for the inner-level optimization of meta-learning, which constitutes a bi-level optimization. Their focus is to streamline the optimization using implicit gradient, while our goal, in contrast, is to use proximal mapping to learn structured data representations.

We note that despite the similarity in titles, [66] differs from our work as it applies proximal mapping in a solver to perform inference in a graphical model, whose cliques are neural networks. The optimization process *happens to* be analogous to a recurrent net, interlaced with proximal maps, and similar analogy has been drawn between the ISTA optimization algorithm and LSTM [67]. We instead use proximal map as a first-class construct/layer in a deep network.

## B  Connecting ProxNet with Meta-learning

In view that ProxNet is a bi-level optimization and the $z$ in (2) may consist of the embeddings of input objects in *mini-batches*, we can interpret ProxNet from a meta-learning perspective. In particular, each mini-batch corresponds to a "task" (or dataset, episode, etc) in the standard meta-learning terminology, and the regularization term corresponds to the task-specific base learner inside each episode of the meta learner. Naturally, the preceding layers serve as the meta-parameters subject to meta-learning. For example, [68–70] used simple metric-based nearest neighbor, [71, 72] optimized

standard learning algorithms iteratively, and [21, 22] leveraged closed-form solutions for base learners. Explicit learning of learner's update rule was investigated in [73–75]. In this sense, ProxNet extends meta-learning to *unsupervised* base learners.

We emphasize that ProxNet only leverages the idea and technique in meta-learning. It is beyond our paper to address existing challenges in meta-learning itself.

**Detailed description**  The conventional meta-learning has a meta-parameter $p$, and each base-learner (for each task) has its own base-parameters $w$. Then by Equation (1) of the paper

Aravind Rajeswaran, Chelsea Finn, Sham Kakade, Sergey Levine. *Meta-Learning with Implicit Gradients*. Neural Information Processing Systems (NeurIPS), 2019,

the bi-level optimization in meta-learning can be set up as ("perf" for "performance"):

$$\min_p \sum_i \text{Test-perf} \left( \arg\min_w \text{Training-perf}(w, p, \mathcal{D}_i^{train}), p, \mathcal{D}_i^{test} \right). \tag{13}$$

Here $\mathcal{D}_i^{train}$ and $\mathcal{D}_i^{test}$ are the training and test data for task $i$, respectively. Now we can establish the one-to-one correspondence between (13) and ProxNet in the context of multiview learning. Please refer to Section 5 for notations, especially Equations (9) and (11).

- $p$: the union of i) the feature extractors $f$ and $g$ for the two views, and ii) the downstream supervised layers. Only the former ($f$ and $g$) is used in the inner training ($\arg\min_w$), which transforms the raw data into the input of the proximal layer.
- $w$: the $U$ and $V$ projection directions used by CCA;
- $\mathcal{D}_i^{train}$: the $i$-th mini-batch $\{(x_i, y_i)\}_{i=1}^n$;
- Training-perf$(w, p, \mathcal{D}_i^{train}) = \min_{P,Q} \frac{\lambda}{2n} \|P - X\|_F^2 + \frac{\lambda}{2n} \|Q - Y\|_F^2 - \text{tr}(U^\top P Q^\top V)$,
  where $X = (f(x_1), \ldots, f(x_n))$ and $Y = (g(y_1), \ldots, g(y_n))$. That is, for any given projection directions $U$ and $V$ (i.e., $w$), what is the minimal denoising objective, which combines the displacement (Frobenius norm) and the CCA objective (correlation between the projections);
- $\mathcal{D}_i^{test}$: the $i$-th mini-batch (same as $\mathcal{D}_i^{train}$);
- Test-perf: pass $\mathcal{D}_i^{test}$ through $f$ and $g$, followed by denoising based on the trained $w = (U, V)$: $\arg\min_{P,Q} \frac{\lambda}{2n} \|P - X\|_F^2 + \frac{\lambda}{2n} \|Q - Y\|_F^2 - \text{tr}(U^\top P Q^\top V)$, and finally apply the supervised layers to measure the test performance.

So ProxNet effectively corresponds to a base-learner of multiview denoising. It extends the common meta-learning practice in two ways:

- the base-learner is unsupervised;
- the training and test performance employ different tasks (denoising versus error).

The latter is quite a valid learning paradigm: the training phase extracts useful representations as parameterized by $U$ and $V$, and then the product ($U$ and $V$) is evaluated on the test data by computing their projections, followed by a supervised loss. Since mini-batch sizes are very small (also intended to keep the optimization efficient), it can be considered as a few-shot learning. Surely the algorithm does not have to be restricted to mini-batches that are drawn iid; different mini-batches can employ bona-fide different learning tasks.

## C  Connecting Proximal Mapping to Kernel Warping

The graph Laplacian on a function $f$ is $\sum_{ij} w_{ij}(f(x_i) - f(x_j))^2$, where $f(x_i) - f(x_j)$ is bounded and linear in $f$. Parameterizing an image as $I(\alpha)$ where $\alpha$ is the degree of rotation/translation/etc, transformation invariance favors a small magnitude of $\frac{\partial}{\partial \alpha}|_{\alpha=0} f(I(\alpha))$, again a bounded linear functional. By Riesz representation theorem, a bounded linear functional can be written as $\langle z_i, f \rangle_{\mathcal{H}}$

for some $z_i \in \mathcal{H}$. We will refer to $z_i$ as an invariance representer, and suppose we have $m$ such invariances.

In order to respect the desired invariances, [36] proposed a warped RKHS $\mathcal{H}^\circ$ consisting of the same functions in the original $\mathcal{H}$, but redefining the norm and the corresponding kernel by

$$\|f\|_{\mathcal{H}^\circ}^2 := \|f\|_{\mathcal{H}}^2 + \sum_{i=1}^m \langle z_i, f \rangle_{\mathcal{H}}^2 \tag{14}$$

This leads to a new RKHS consisting of the same set of functions as $\mathcal{H}$, but its inner product warped into

$$\langle f, g \rangle_{\mathcal{H}^\circ} := \langle f, g \rangle_{\mathcal{H}} + \sum_{i=1}^m \langle f, z_i \rangle_{\mathcal{H}} \langle g, z_i \rangle_{\mathcal{H}}, \tag{15}$$

and its kernel is warped into

$$k^\circ(x_1, x_2) = k(x_1, x_2) - z(x_1)^\top K_Z z(x_2), \tag{16}$$

where $z(x) = (z_1(x), \ldots, z_m(x))^\top$. Then replacing $k(x, \cdot)$ by $k^\circ(x, \cdot)$ results in a new invariant representation. Such a warping can be applied to all layers in, e.g., deep convolutional kernel networks [CKNs, 76], instilling invariance with respect to preceding layer's output.

The major limitation of this method, however, is that the invariances have to be modeled by the square of a linear form — $\langle z_i, f \rangle_{\mathcal{H}}^2$ — in order to make $\|f\|_{\mathcal{H}}^2 + \sum_{i=1}^m \langle z_i, f \rangle_{\mathcal{H}}^2$ a norm square, precluding many interesting invariances such as total variation $f \mapsto \int |f'(x)| \, dx$.

Interestingly, this can be achieved by simply reformulating kernel warping as proximal mapping. To this end, recall that a Euclidean embedding maps $f \in \mathcal{H}$ to a real vector $\tilde{f}$, such that $\langle \tilde{f}, \tilde{h} \rangle \approx \langle f, h \rangle_{\mathcal{H}}$ for all $f, h \in \mathcal{H}$. A commonly used formula for embedding is the Nyström approximation [77]. Using $p$ samples $W := \{\omega_i\}_{i=1}^p$ drawn i.i.d. from $\mathcal{X}$, we derive an embedding of $f \in \mathcal{H}$ as follows, ensuring that $\langle \tilde{f}, \tilde{h} \rangle \approx \langle f, h \rangle_{\mathcal{H}}$ for all $f, h \in \mathcal{H}$:

$$\tilde{f} := K_W^{-1/2} f_W, \quad \text{where} \quad K_W := (k(\omega_i, \omega_j))_{ij} \in \mathbb{R}^{p \times p}, \quad f_W := (f(\omega_1), \ldots, f(\omega_p))^\top \in \mathbb{R}^p.$$

Let $\tilde{\varphi}(x)$ be the embedding of $k(x, \cdot)$, and $\tilde{Z} := (\tilde{z}_1, \ldots, \tilde{z}_m)$ where $\tilde{z}_i$ is the embedding of the invariance representer $z_i$. Then [36] showed that the Euclidean embedding of $k^\circ(x, \cdot)$ can be written as

$$(I + \tilde{Z}\tilde{Z}^\top)^{-1/2} \tilde{\varphi}(x). \tag{17}$$

Now to apply proximal map, it is natural to set $L(f) = \frac{1}{2} \sum_{i=1}^m \langle z_i, f \rangle_{\mathcal{H}}^2$ to enforce invariance. Then the proximal map $\mathsf{P}_L(k(x, \cdot))$ for the representer $k(x, \cdot)$ with $\lambda = 1$ is

$$\mathsf{P}_L(k(x, \cdot)) = \arg\min_{f \in \mathcal{H}} \left\{ L(f) + \tfrac{1}{2} \|f - k(x, \cdot)\|_{\mathcal{H}}^2 \right\} \tag{18}$$

$$= \arg\min_{f \in \mathcal{H}} \left\{ \tfrac{1}{2} \sum_{i=1}^m \langle z_i, f \rangle_{\mathcal{H}}^2 + \tfrac{1}{2} \|f - k(x, \cdot)\|_{\mathcal{H}}^2 \right\} \tag{19}$$

$$= (I + ZZ^\top)^{-1} k(x, \cdot). \tag{20}$$

Its Euclidean embedding can be obtained by replacing $z_i$ with $\tilde{z}_i$, and $k(x, \cdot)$ with $\tilde{\varphi}(x)$:

$$\arg\min_{v \in \mathbb{R}^p} \left\{ \tfrac{1}{2} \sum_{i=1}^m \langle \tilde{z}_i, v \rangle^2 + \tfrac{1}{2} \|v - \tilde{\varphi}(x)\|^2 \right\} = (I + \tilde{Z}\tilde{Z}^\top)^{-1} \tilde{\varphi}(x). \tag{21}$$

This is almost the same as that from kernel warping in (17), except for the exponent on $I + \tilde{Z}\tilde{Z}^\top$. In practice, we observed that it led to little difference, and the result of proximal mapping using Gaussian kernel and flat-gradient invariance is shown in Figure 1. That is, $L(f) = \frac{1}{2} \sum_i \|\nabla f(x_i)\|^2$. Trivially, CKNs can now leverage nonlinear invariances such as total variation by using a nonlinear regularizer $L$ in (18).

## D Simulations for Connecting Proximal Mapping to Dropout

We now use the two-moon dataset to verify that only small differences arise if dropout is implemented by proximal mapping in Section 3, as opposed to the adaptive regularization in (6). Suppose the

$i$-th training examples is $x_i \in \mathbb{R}^d$ with label $y_i \in \{-1, 1\}$. The $j$-th feature of $x_i$ is denoted as $x_{ij}$. Employing logistic loss, the adaptive regularization view of dropout by [37] can be written as

$$\beta_* := \min_{\beta \in \mathbb{R}^d} \left\{ \frac{1}{n} \sum_{i=1}^n \log(1 + \exp(-y_i \beta^\top x_i)) + \mu \sum_j a_j \beta_j^2 \right\}, \tag{22}$$

where $a_j = \frac{1}{n} \sum_{i=1}^n p_i(1-p_i)x_{ij}^2, \quad p_i = (1 + \exp(-\beta^\top x_i))^{-1}$.

Our proximal map is defined as

$$\mathsf{P}_R(x) = \arg \min_{z \in \mathbb{R}^d} \left\{ \frac{\lambda}{2} \|z - x\|_2^2 + \sum_j b_j z_j^2 \right\}, \tag{23}$$

where $b_j = \frac{1}{n} \sum_{i=1}^n q_i(1-q_i)x_{ij}^2, \quad q_i = (1 + \exp(-z^\top x_i))^{-1}$.

And the output layer is trained by

$$\alpha_* := \min_{\alpha \in \mathbb{R}^d} \left\{ \frac{1}{n} \sum_{i=1}^n \log(1 + \exp(-y_i \alpha^\top \mathsf{P}_R(x_i)) + c\|\alpha\|^2 \right\}. \tag{24}$$

To demonstrate that the two methods yield similar discriminant values, we produce a scatter plot of $\alpha_*^\top P_R(x_i)$ (for proximal mapping) versus $\beta_*^\top x_i$ (for dropout). Figure 6 shows the result for two example settings. Clearly, the two methods produce similar discriminant values for all training examples. The Matlab code is also available on GitHub.

(a) $\lambda = 0.5$, $\mu = 0.1$, and $c = 0.2\lambda^2\mu$    (b) $\mu = 0.1$, $\lambda = 0.1$, and $c = 15\lambda^2\mu$

Figure 6: Scatter plot of $\alpha_*^\top P_R(x_i)$ ($y$-axis for proximal mapping) versus $\beta_*^\top x_i$ ($x$-axis for dropout)

For optimize (24), we simply invoked fminunc without providing any gradient subroutine. That is, fminunc was98 left to choose its own solver which typically utilizes its own finite difference routine. The result looks good and efficient for this dataset.

# E    ProxNet for Multiview Learning

Most multiview learning algorithms are based on CCA, which most commonly involves only two views. It is in fact not hard to extend it to more than two views. For example, [78] proposed that given $J$ centered views $X_j \in \mathbb{R}^{N \times d_j}$ for $j \in [J]$, where $N$ is the number of training examples and $d_j$ is the dimensionality of the $j$-th view, the generalized CCA (GCCA) can be written as the following optimization problem

$$L(\{X_j\}_{j=1}^J) := \min \sum_{j=1}^J \|G - X_j U_j\|_F^2, \tag{25}$$

where $G \in \mathbb{R}^{N \times r}$, $U_j \in \mathbb{R}^{d_j \times r}$, $G^\top G = I$. Intuitively, it finds a linear transformation $U_j$ for each view, so that all views can be transformed to a similar core $G$. Furthermore, $G$ needs to be orthonormal, to avoid mode collapse. The optimal value, denoted as $L(\{X_j\})$, will be used as the $L$ function in (11).

Furthermore, given $\{X_j\}$, (25) can be optimized efficiently in closed form based on generalized eigenvalues [78–80]. Based on the optimal solution of $G$ and $\{U_j\}$, the derivative of $L(\{X_j\})$ in $\{X_j\}$ can be directly computed by Danskin's theorem.

## F  Backpropagation Through Time for Adversarial LSTM

To concentrate on backpropagation, we assume that the ultimate objective $J$ only depends only on the output of the last time step $T$, i.e., $h_T$. Extension can be easily made to the case where each step also contributes to the overall loss. From the final layer, we get $\frac{\partial J}{\partial h_T}$. Then we can get $\frac{\partial J}{\partial h_{T-1}}$ and $\frac{\partial J}{\partial c_{T-1}}$ as in the standard LSTM ($G_T$ in the final layer can be ignored and $\frac{\partial J}{\partial c_T} = 0$). In order to compute the derivatives with respect to the weights $W$ in the LSTMs, we need to recursively compute $\frac{\partial J}{\partial h_{t-1}}$ and $\frac{\partial J}{\partial c_{t-1}}$, given $\frac{\partial J}{\partial h_t}$ and $\frac{\partial J}{\partial c_t}$. Once they are available, then

$$
\frac{\partial J}{\partial W} = \sum_{t=1}^{T} \left\{ \underbrace{\frac{\partial J}{\partial h_t}}_{\text{by (27)}} \underbrace{\frac{\partial}{\partial W} h_t(c_{t-1}, h_{t-1}, x_t)}_{\text{standard LSTM}} + \underbrace{\frac{\partial J}{\partial c_t}}_{\text{by (30)}} \underbrace{\frac{\partial}{\partial W} c_t(c_{t-1}, h_{t-1}, x_t)}_{\text{standard LSTM}} \right\}, \tag{26}
$$

where the two $\frac{\partial}{\partial W}$ on the right-hand side are identical to the standard operations in LSTMs. Here we use the Jacobian matrix arrangement for partial derivatives, i.e., if $f$ maps from $\mathbb{R}^n$ to $\mathbb{R}^m$, then $\frac{\partial f(x)}{\partial x} \in \mathbb{R}^{m \times n}$.

Given $\frac{\partial J}{\partial c_t}$, we can first compute $\frac{\partial J}{\partial s_t}$ and $\frac{\partial J}{\partial G_t}$ based on the proximal map, and the details will be provided in Section F.1. Given their values, we now compute $\frac{\partial J}{\partial h_{t-1}}$ and $\frac{\partial J}{\partial c_{t-1}}$. Firstly,

$$
\frac{\partial J}{\partial h_{t-1}} = \underbrace{\frac{\partial J}{\partial h_t}}_{\text{by recursion}} \underbrace{\frac{\partial h_t}{\partial h_{t-1}}}_{\text{std LSTM}} + \underbrace{\frac{\partial J}{\partial G_t} \frac{\partial G_t}{\partial h_{t-1}}}_{\text{by (28)}} + \underbrace{\frac{\partial J}{\partial s_t}}_{\text{by (37)}} \underbrace{\frac{\partial s_t}{\partial h_{t-1}}}_{\text{std LSTM}}. \tag{27}
$$

The terms $\frac{\partial h_t}{\partial h_{t-1}}$ and $\frac{\partial s_t}{\partial h_{t-1}}$ are identical to the operations in the standard LSTM. The only remaining term is in fact a directional second-order derivative, where the direction $\frac{\partial J}{\partial G_t}$ can be computed from from (47):

$$
\frac{\partial J}{\partial G_t} \frac{\partial G_t}{\partial h_{t-1}} = \frac{\partial J}{\partial G_t} \frac{\partial^2}{\partial x_t \partial h_{t-1}} s_t(c_{t-1}, h_{t-1}, x_t) \tag{28}
$$

$$
= \frac{\partial}{\partial h_{t-1}} \left\langle \underbrace{\frac{\partial J}{\partial G_t}}_{\text{by (47)}}, \frac{\partial}{\partial x_t} s_t(c_{t-1}, h_{t-1}, x_t) \right\rangle. \tag{29}
$$

Such computations are well supported in most deep learning packages, such as PyTorch. Secondly,

$$
\frac{\partial J}{\partial c_{t-1}} = \underbrace{\frac{\partial J}{\partial h_t}}_{\text{by recursion}} \underbrace{\frac{\partial h_t}{\partial c_{t-1}}}_{\text{std LSTM}} + \underbrace{\frac{\partial J}{\partial G_t} \frac{\partial G_t}{\partial c_{t-1}}}_{\text{by (31)}} + \underbrace{\frac{\partial J}{\partial s_t}}_{\text{by (37)}} \underbrace{\frac{\partial s_t}{\partial c_{t-1}}}_{\text{std LSTM}}. \tag{30}
$$

The terms $\frac{\partial h_t}{\partial c_{t-1}}$ and $\frac{\partial s_t}{\partial c_{t-1}}$ are identical to the operations in the standard LSTM. The only remaining term is in fact a directional second-order derivative:

$$
\frac{\partial J}{\partial G_t} \frac{\partial G_t}{\partial c_{t-1}} = \frac{\partial J}{\partial G_t} \frac{\partial^2}{\partial x_t \partial c_{t-1}} s_t(c_{t-1}, h_{t-1}, x_t) \tag{31}
$$

$$
= \frac{\partial}{\partial c_{t-1}} \left\langle \underbrace{\frac{\partial J}{\partial G_t}}_{\text{by (47)}}, \frac{\partial}{\partial x_t} s_t(c_{t-1}, h_{t-1}, x_t) \right\rangle. \tag{32}
$$

### F.1 Gradient Derivation for the Proximal Map

We now compute the derivatives involved in the proximal operator, namely $\frac{\partial J}{\partial s_t}$ and $\frac{\partial J}{\partial G_t}$. For clarify, let us omit the step index $t$, set $\delta = \sqrt{\lambda}$ without loss of generality, and denote

$$J = f(c), \quad \text{where} \quad c := c(G, s) := (I + GG^\top)^{-1}s. \tag{33}$$

We first compute $\partial J/\partial s$ which is easier.

$$\Delta J := f(c(G, s + \Delta s)) - f(c(G, s)) \tag{34}$$

$$= \nabla f(c)^\top (c(G, s + \Delta s) - c(G, s)) + o(\|\Delta s\|) \tag{35}$$

$$= \nabla f(c)^\top (I + GG^\top)^{-1} \Delta s + o(\|\Delta s\|). \tag{36}$$

Therefore,

$$\frac{\partial J}{\partial s} = \nabla f(c)^\top (I + GG^\top)^{-1}. \tag{37}$$

We now move on to $\partial J/\partial G$. Notice

$$\Delta J := f(c(G + \Delta G, s)) - f(c(G, s)) \tag{38}$$

$$= \nabla f(c)^\top (c(G + \Delta G, s) - c(G, s)) + o(\|\Delta G\|). \tag{39}$$

Since

$$c(G + \Delta G, s) = (I + (G + \Delta G)(G + \Delta G)^\top)^{-1}s \tag{40}$$

$$= \left[ (I + GG^\top)^{\frac{1}{2}} \left( I + (I + GG^\top)^{-\frac{1}{2}} (\Delta G G^\top + G \Delta G^\top)(I + GG^\top)^{-\frac{1}{2}} \right) (I + GG^\top)^{\frac{1}{2}} \right]^{-1} s \tag{41}$$

$$= (I + GG^\top)^{-\frac{1}{2}} \left( I - (I + GG^\top)^{-\frac{1}{2}} (\Delta G G^\top + G \Delta G^\top)(I + GG^\top)^{-\frac{1}{2}} + o(\|\Delta G\|) \right) (I + GG^\top)^{-\frac{1}{2}} s \tag{42}$$

$$= c(G, s) - (I + GG^\top)^{-1} (\Delta G G^\top + G \Delta G^\top)(I + GG^\top)^{-1} s + o(\|\Delta G\|), \tag{43}$$

we can finally obtain

$$\Delta J = -\nabla f(c)^\top (I + GG^\top)^{-1} (\Delta G G^\top + G \Delta G^\top)(I + GG^\top)^{-1} s + o(\|\Delta G\|) \tag{44}$$

$$= -\operatorname{tr}\left( \Delta G^\top (I + GG^\top)^{-1} \left( \nabla f(c)s^\top + s\nabla f(c)^\top \right) (I + GG^\top)^{-1} G \right) + o(\|\Delta G\|). \tag{45}$$

So in conclusion,

$$\frac{\partial J}{\partial G} = -(I + GG^\top)^{-1} \left( \nabla f(c)s^\top + s\nabla f(c)^\top \right) (I + GG^\top)^{-1} G \tag{46}$$

$$= -(ac^\top + ca^\top)G, \quad \text{where} \quad a = (I + GG^\top)^{-1}\nabla f(c). \tag{47}$$

## G   Detailed Experimental Result

**All code and data are available anonymously, with no tracing, at**

```
https://github.com/learndeep2019/ProxNet.
```

We will demonstrate the effectiveness of ProxNet on several multi-view learning tasks including image classification, speech recognition, and crosslingual word embedding. Four baseline methods were selected for comparison in multi-view learning:

- **Vanilla** model: a network is trained for each view without CCA regularization, and the output of the two views were combined by averaging their logits for supervised tasks. The network is trained in an end-to-end manner.
- **DCCA** [10]: a network is trained to learn a pair of highly-correlated representations for the two views, which are then used for training the subsequent supervised task. The whole model is trained in a disjoint manner.

- **DCCAE** [9]: same as DCCA, except that it trains an extra decoder to enforce that the learned representations can well reconstruct the input.
- **RRM**: connect the code/output of DCCA with a supervised classifier, and train it with the encoder in an end-to-end fashion. It also resembles ProxNet, except that the regularizer $L(X, Y)$ is moved from the proximal layer to the overall objective as in (1) (i.e., no more proximal mapping).

## G.1 Multiview Supervised Learning: image recognition with sketch and photo

**Dataset.** We first evaluated ProxNet on a large scale sketch-photo paired database – Sketchy which consists of 12,500 photos and 75,471 hand-drawn sketches of objects from 125 classes. Each sample from sketch and photo is $256 \times 256$ colored natural images. To demonstrate the robustness of our method, we varied the number of classes over $\{20, 50, 100, 125\}$ by sampling a subset of classes from the original dataset. For each class, there are 100 sketch-photo pairs. We randomly selected 80 pairs of photo and sketch from each same class to form the training set, and then used the remaining 20 pairs for testing.

**Implementation detail.** Our implementation was based on PyTorch and all training was conducted on one NVIDIA GeForce 2080Ti GPU.

For all methods, we used ResNet-18 as the feature extractor. In ProxNet, the feature extractor immediately followed by a proximal layer which has input and output dimension $d = 20$. Then a classifier which has three fully-connected layer each having 512 units was trained on the outputs of proximal layer. The final output layer has multiple softmax units that each corresponds to the output classes. At training time, we employed an adaptive trade-off parameter $\lambda_t = (1 + kt)\alpha_0$, where $k = 0.5$ and $\alpha_0 = 0.1$. RRM used the same architecture as ProxNet, except that, instead of using the proximal layer, RRM moves the CCA objective (i.e., the regularizer $L(X, Y)$) to the overall objective to promote the correlation between the two views' hidden representation.

Since the Vanilla model does not promote correlation between views, it can be adapted from RRM model by removing the regularizer from the overall objective. [9, 10] trained DCCA and DCCAE in two separate steps instead of end-to-end. The first step learned an encoder (and decoder for DCCAE) to optimize the CCA objective, and the second step trained a supervised classifier based on the code. In our experiment, their encoders employed the same architecture as the feature extractors of ProxNet and other baselines, i.e., ResNet-18. For DCCAE, we built a CNN-based decoder to reconstruct the inputs.

For all methods, the loss was evaluated on the averaged logits at training time in order to be consistent with how predictions were made at test time.

We used the Ray Tune library to select the hyper-parameters for all methods, and the selected parameters are summarized here:

Table 5: Hyper-parameters for all methods on the Sketchy dataset

| Hyper-parameters | Vanilla | DCCA | DCCAE | RRM | ProxNet |
|---|---|---|---|---|---|
| Dimension $d$ | 15 | 22 | 19 | 20 | 20 |
| Optimizer | Adam | Adam | Adam | Adam | Adam |
| Learning rate | 0.0012 | 0.0010 | 0.0009 | 0.0011 | 0.0011 |
| Weight decay | $10^{-4}$ | $10^{-4}$ | $10^{-4}$ | $10^{-4}$ | $10^{-4}$ |

The accuracy of all methods saturates after the mini-batch size goes above 100. So we just used 100 for all methods to keep training efficient.

## G.2 Audio-Visual Speech Recognition

**Dataset.** In this task, we aim to use learned features for speaker-independent phonetic recognition. We experimented on the Wisconsin X-ray Micro-Beam Database (XRMB) corpus which consists of simultaneously recorded speech and articulatory measurements from 47 American English speakers

and 2357 utterances. The two raw-input views are acoustic features (39D mel frequency cepstral coefficients (MFCCs) and their first and sencond derivatives) and articulatory features (16D horizontal/vertical displacement of 8 pellets attached to several parts of the vocal tract). Along with the multi-view data there are phonetic labels available for classification. To simulate the real-life scenarios and improve the model's robustness to noise, the acoustic features of a given speaker are corrupted by mixing with $\{0.2, 0.5, 0.8\}$ level of another random picked speaker's acoustic features. The XRMB speakers were partitioned into disjoint sets of 35/12 speakers for training and testing respectively.

**Implementation detail.**    In [9], to incorporate contexts information, the inputs are concatenated over a $W$-frame window centered at each frame, giving $39 \times W$ and $16 \times W$ feature dimensions for each of the two views respectively. Although this delicately construed inputs freed the encoder/feature extractor from considering the time dependency within frames, we prefer a refined modeling of the sequential structure. Therefore, instead of concatenating features for each $W$-frame window followed by a fully connected network as in [45], we implemented, for all algorithms under consideration, a 2-layer LSTM with hidden size 256. The output of LSTM was passed through a fully connected layer, projecting to a $K$-dimensional subspace. This feature extractor significantly improved the performance of all methods.

The supervised predictor was implemented by a fully connected network of 2 hidden layers each having 256 ReLU units, and a linear output layer of 41 log-softmax units. We used Pytorch's built-in function Connectionist Temporal Classification (CTC) loss [44] with greedy search as the phone recognizer. Again, all methods shared the same architecture of supervised predictor.

Both RRM and Vanilla were trained in the same way as for the Sketchy dataset in Section G.1. To train ProxNet, we employed an adaptive trade-off parameter $\lambda_t = (1 + kt)\alpha_0$, where $k = 1$ and $\alpha_0 = 0.5$. DCCA and DCCAE performed poorly if only the learned code/features were used for phonetic recognition. Therefore, we followed [9] and concatenated them with the original features (39D and 16D for the acoustic and articulatory views, respectively), based on which a CTC-based recognizer is trained. This improved the PER performance of DCCA and DCCAE significantly.

In the logit averaging mode, all methods were trained with a loss applied to the averaged logits. This is the same as Section G.1. In the acoustic mode, however, a loss is applied to each view at training time based on the ground truth label. These are both consistent with how predictions are made at test time.

Here we intentionally used $K$ instead of $d$ to denote the hidden dimension. This is to avoid confusion because LSTM is used as in Figure 4. For a mini-batch of size $m$ where each sequence has length $s$, the input of the proximal layer is in fact $m \cdot s$ examples of $K$ dimensional. Although $m \cdot s$ may result in a large number, the proximal mapping can still be solved efficiently because we were able to use a larger value of $\lambda$ in this dataset. In addition, the computational cost for SVD on an $ms$-by-$K$ matrix is $O(msK^2)$ when $K \leq ms$. Since we used $K = 20$, the quadratic dependency on $K$ did not create a computational challenge in practice.

As in the Skytch dataset, we used the Ray Tune library to select the hyper-parameters, and the selected parameters are summarized here:

Table 6: Hyper-parameters for all methods on XRMB

| Hyper-parameters | Vanilla | DCCA | DCCAE | RRM | ProxNet |
|---|---|---|---|---|---|
| Dimension $K$ | 12 | 20 | 20 | 18 | 20 |
| Optimizer | Adam | Adam | Adam | Adam | Adam |
| Learning rate | 0.0009 | 0.0011 | 0.0010 | 0.0013 | 0.0010 |
| Weight decay | 0.0005 | 0.0005 | 0.0005 | 0.0005 | 0.0005 |

Line 250 made an inaccurate description of how we tuned $K$: "The dimension of subspace was tuned in $\{10, 20, 30, 50\}$, and the sequence length was tuned in $\{250, 500, 1000\}$ for all algorithms". This was the setting in our preliminary experiment. The Ray Tune library indeed allowed us to later search all parameters in a continuous space, and so the $K$ values in Table 6 can be 12 or 18.

We eventually set the sequence length to $s = 1000$ for all methods, because it consistently produced the best result, which is not surprising because longer sequences can preserve more structure. However, the PER saturated after the length rose beyond 1000.

Similarly, the PER of all methods leveled off after the mini-batch size grew above 32. So we just used $m = 32$ for all methods to keep training efficient.

**Evaluation.**   For all experiments, we report the Phone error rates (PERs) which is defined as $PER = (S + D + I)/N$, where $S$ is the number of substitutions, $D$ is the number of deletions, $I$ is the number of insertions to get from the reference to the hypothesis, and $N$ is the number of phonetics in the reference. The PERs obtained by different methods are given in Table 2.

### G.3   Crosslingual/Multilingual Word Embedding

In this task, we learned representation of English and German words from the paired (English, German) word embeddings for improved semantic similarity.

**Dataset.**   We first built a parallel vocabulary of English and German from the parallel news commentary corpora [WMT 2012-2018 46] using the word alignment method from [47, 48]. Then we selected 36K English-German word pairs, in descending order of frequency, for training. Based on the vocabulary we also built a bilingual dictionary for testing, where each English word $x_i$ is matched with the (unique) German word $y_i$ that has been most frequently aligned to $x_i$. Unlike the setup in [53] and [9], where word embeddings are trained via Latent Semantic Analysis (LSA) using parallel corpora, we used the pretrained monolingual 300-dimensional word embedding from [50] and [49] as the raw word embeddings ($x_i$ and $y_i$).

To evaluate the quality of learned word representation, we experimented on two different benchmarks that have been widely used to measure word similarity [51, 52]. Multilingual WS353 contains 353 pairs of English words, and their translations to German, Italian and Russian, that have been assigned similarity ratings by humans. It was further split into Multilingual WS-SIM and Multilingual WS-REL which measure the similarity and relatedness between word pairs respectively. Multilingual SimLex999 is a similarity-focused dataset consisting of 666 noun pairs, 222 verb pairs, 111 adjective pairs, and their translations from English to German, Italian and Russian.

**Baselines.**   We compared our method with the monolingual word embedding (baseline method) from fastText to show that ProxNet learned a good word representation through the proximal layer. Since our method is mainly based on CCA, we also chose three competitive CCA-based models for comparison, including:

- linearCCA [53], which applied a linear projection on the two languages' word embedding and then projected them into a common vector space such that aligned word pairs should be maximally correlated.
- DCCA [81], which, instead of learning linear transformations with CCA, learned nonlinear transformations of two languages' embedding that are highly correlated.
- DCCAE [9], which noted that there is useful information in the original inputs that is not correlated across views. Therefore, they not only projected the original embedding into subspace, but also reconstructed the inputs from the latent representation.
- CL-DEPEMB [54], a novel cross-lingual word representation model which injects syntactic information through dependency-based contexts into a shared cross-lingual word vector space.

**Implementation detail.**   We first used the fastText model to embed the 36K English-German word pairs into vectors. Then we normalized each vector to unit $\ell_2$ norm and removed the per-dimension mean and standard deviation of the training pairs.

To build an end-to-end model, we followed the same intuition as DCCAE but instead of using the latent representation from the encoder to reconstruct the inputs, we used the outputs of proximal layer, which is a proximal approximation of latent representation from the encoder, to do the reconstruction. That is, the input reconstruction error was used as the ultimate objective.

Table 7: Summary of datasets for adversarial LSTM training

| Dataset | Training | Test | Median length | Attributes | Classes |
|---|---|---|---|---|---|
| JV | 225 | 370 | 15 | 12 | 9 |
| HAR | 6,127 | 2,974 | 128 | 9 | 6 |
| AD | 5,500 | 2,200 | 39 | 13 | 10 |
| IMDB | 25,000 | 25,000 | 239 | - | 2 |

We implemented the encoder (feature mapping $f$ and $g$) by using multilayer perceptrons with ReLU activation and the decoder by using a symmetric architecture of encoder. We tuned the hidden dimension $h$ for $f$ and $g$ among $\{0.1, 0.3, 0.5, 0.7, 0.9\} \times 300$, the regularization parameter $\lambda$ from $\{0.001, 0.01, 0.1, 1, 10\}$, and the depth and layer width from 1 to 4 and $\{256, 512, 1024, 2048\}$, respectively. For optimization, we used SGD with momentum 0.99, a weight decay of 0.0005, and a learning rate 0.1 which was divided by 10 after 100 and 200 epochs.

At test time, for numerical stability, we combined the word vectors from bilingual dictionary and the test set to build paired vocabulary for each language. We applied the same data preprocessing (normalize to unit norm, remove the mean/standard deviation of the training set) on test vocabularies (English and German word vectors). Then we feed paired test vocabularies into the models and obtained the word representation of the test data. We projected the output of the proximal layer to the subspace where each paired word representation was maximally correlated. The projection matrices were calculated from the 36K training set through the standard CCA method. We computed the cosine similarity between the final word vectors in each pair, ordered the pairs by similarity, and computed the Spearman's correlation between the model's ranking and human's ranking.

### G.4 Adversarial Training in Recurrent Neural Network

Here we include more details on the experiment of adversarial training in recurrent neural network as described in Section 6.4.

**Datasets.** To demonstrate the effectiveness of using proximal mapping, we tested on four different sequence datasets. The Janpanese Vowels dataset [JV 55] contains time series data where nine male speakers uttered Japanese Vowels successively, and the task is to classify speakers. The Human Activity Recognition dataset [HAR 56] is used to classify a person's activity (sitting, walking, etc.) based on a trace of their movement using sensors. The Arabic Digits dataset [AD, 57] contains time series corresponding to spoken Arabic digits by native speakers, and the task is to classify digits. IMDB [58] is a standard movie review dataset for sentiment classification. Details of the datasets are summarized in Table 7. The - is because IMDB is a text dataset, for which a 256-dimensional word embedding is learned.

**Preprocessing.** Normalization was the only preprocessing applied to all datasets. For those datasets that contain variable-length sequences, zero-padding was used to make all sequences have the same length as the longest sequence in a mini-batch. To reduce the effect of padding, we first sorted all sequences by length (except the IMDB dataset), so that sequences with similar length were assigned to the same mini-batch.

**Baseline models.** To show the impact of applying proximal mapping on LSTM, we compared our method with two baselines. For JV, HAR and AD datasets, the base model structure was composed of a CNN layer, followed by an LSTM layer and a fully-connected layer. The CNN layer was constructed with kernel size 3, 8, 3 and contained 32, 64, 64 filters for JV, HAR, AD respectively. For the LSTM layer, the number of hidden units used in these three datasets are 64, 128, 64, respectively. This architecture was denoted as LSTM in Table 4. For IMDB, following [40], the basic model consisted of a word embedding layer with dimension 256, a single-layer LSTM with 1024 hidden units, and a hidden dense layer of dimension 30.

On top of this basic LSTM structure, we compared two different adversarial training methods. AdvLSTM is the adversarial training method in [40], which we reimplemented in PyTorch, and perturbation was added to the input of each LSTM layer. ProxLSTM denotes our method described

in Section 4, where the LSTM cell in the basic structure was replaced by our ProxLSTM cell. LSTM and AdvLSTM here correspond to "Baseline" and "Adversarial" in [40] respectively.

**Training.**    For the JV, HAR, AD datasets, we first trained the baseline LSTM to convergence, and then applied AdvLSTM and ProxLSTM as fine tunning, where ADAM was used with learning rate $10^{-3}$ and weight decay $10^{-4}$. For IMDB, we first trained LSTM and AdvLSTM by following the settings in [40], with an ADAM optimizer of learning rate $5 \cdot 10^{-4}$ and exponential decay 0.9998. Then the result of AdvLSTM was used to initialize the weights of ProxLSTM. All settings were evaluated 10 times to report the mean and standard deviation.

**Results.**    The test accuracies were summarized in Table 4. Clearly, adversarial training improves the performance, and ProxLSTM even promotes the performance more than AdvLSTM. Figure 5 illustrates the t-SNE embedding of extracted features from the last time step's hidden state of HAR test set. Although ProxLSTM only improves upon AdvLSTM marginally in test accuracy, Figure 5 shows the embedded features from ProxLSTM cluster more compactly than those of AdvLSTM (e.g. the yellow class). The t-SNE plot of other datasets are available in Figures 7, 8 and 9. This further indicates that ProxLSTM can learn better latent representation than AdvLSTM by applying proximal mapping.

Figure 7: t-SNE embedding of the JV dataset

Figure 8: t-SNE embedding of the AD dataset

Figure 9: t-SNE embedding of the IMDB dataset

## Footnotes

[1] Although the original paper only detailed on quadratic optimization mainly for the efficient GPU implementation, it is conceptually applicable to general nonlinear optimization. Such extensions have been achieved in [15].