[Reviews · NeurIPS 2020]

Review 1

Summary and Contributions: This paper investigates a novel regularization technique based on introducing a proximal mapping transformation into a deep neural network. The authors first show an example of a proximal mapping being used in the optimization of a shallow model. Here a conventional regularized objective is replaced with an objective that uses proximal mapping instead (explicit rather than implicit regularization; for a positive semi-definite quadratic regularizer this internal optimization for the proximal mapping can be done in a closed form). The proximal mapping approach is then applied to robust learning in LSTMs and for multiview learning. I thank the authors for their rebuttal. Having read it and the other reviews, I keep my recommendation as "accept".

Strengths: After briefly reviewing theoretical derivations, I think that they are sound and well grounded. The method itself is interesting and well-justified and I believe it to be both novel and significant. In my opinion, the proposed regularization technique may potentially find many interesting applications in the field and I am looking forward to seeing them. Even though I am not particularly well familiar with multiview learning, presented results (both ProxLSTM and multiview learning) appear to be sound and promising. Since this article both presents a novel idea and shows several promising deep learning applications, I believe it to be of relevance for the NeurIPS community.

Weaknesses: While the approach itself and specific applications to LSTMs and ProxNet look promising, the paper does not explore more straightforward applications to per-layer regularization in deep neural networks (thus laying a foundation for more complex applications), potentially expanding a toy example shown in Figure 1. Mentioning this seemingly fundamental application, the authors do not study it just saying that it is straightforward to implement. This makes the reader wonder whether this method can actually show promising results in this setting (akin those hinted at in Figure 1).

Correctness: I did not check most of the derivations in the supplementary materials, but I read them quickly and I did not see any immediate issues, which makes me believe that to the best of my knowledge, most claims appear to be correct. Similarly, I did not read the source code and do not know all of the details of the implementations, but the overall empirical methodology outlined in Section 6 is sound.

Clarity: The paper is well written. I assume Eq. (4) should read argmin instead of argmax, but I did not see any other obvious issues. The case of a general R in Section 3 was a little difficult to follow without referring to supplementary materials (same also applies to Section 5.1), but otherwise the text is understandable even if a little overloaded.

Relation to Prior Work: While I am not thoroughly familiar with related work, after a short literature overview, I believe this work to be novel. While there are numerous examples of proximal mapping applications in the field of deep learning (including those in optimization methods), specific ideas and designs developed in this work are novel to the best of my knowledge. The paper itself provides a very brief overview of related literature with a much more detailed discussion presented in Appendix A.

Reproducibility: No

Additional Feedback: The optimization of objective (22) with the proximal map (21) is not discussed in detail in Appendix D. Despite the claim that the Matlab code is available on GitHub, I could not find it in the GitHub repository. This led to me reply "No" to the reproducibility question.


Review 2

Summary and Contributions: Thanks for the clarifications in the rebuttal. This is a good paper and I've increased my score from a marginal accept to an accept. --- This paper studies model regularization via proximal operators. They show how these relate to regularized risk minimization in general and present how to practice use these in theory. As examples they instantiate an LSTM layer that uses a proximal mapping to encourage invariance to perturbations to the inputs, and study proximal operations for multiview learning.

Strengths: Better-understanding model regularization is an important topic to address and this paper presents a nice collection of general ideas paired with instantiations in non-trivial domains. Their regularized models almost always surpass the baseline methods on Sketchy, XRMB, HAR, and other sequential datasets. The interpretation of existing operations as proximal mappings as discussed in S2 is insightful.

Weaknesses: Proximal operators provide an extremely large function class to to optimize and it seems like this flexibility could hurt the model's performance if they are not instantiated correctly for the problem and model that are being considered. I do not have much intuition on how the ProxNet improvements on the datasets they consider compare to other regularization approaches considered on the tasks beyond the RRM baselines presented in this paper.

Correctness: I see no correctness errors in this paper

Clarity: The paper is clear and well-written in most parts. The one area I did not find clear is in the experiments, it's difficult to understand exactly how the proximal operator is instantiated and paramaterized for every experiment.

Relation to Prior Work: The paper already discusses a significant amount of related literature and methods and discusses how existing regularization approaches can be interpreted as proximal operators. One area that may be worth adding is the existing use cases of proximal operators for regularizing other optimization problems, such as: Meinhardt, T., Moller, M., Hazirbas, C., & Cremers, D. Learning proximal operators: Using denoising networks for regularizing inverse imaging problems. CVPR 2017.

Reproducibility: Yes

Additional Feedback: The finite-difference derivative approximation in L178 seems like it can cause instability during learning. Does care need to be taken to properly setting up the proximal operator and setting the \epsilon term to ensure the derivative approximation is well-behaved?


Review 3

Summary and Contributions: The authors present ProxNet, an approach to regularized optimization that replaces regularization terms in the loss with regularization steps in the model. They show that many (data-independent) regularizers, as well as "data-dependent regularizers" like the regularization of multiple embeddings towards each other in multi-view learning, can be reformulated with ProxNet. They demonstrate substantially improved performance over regularization-in-the-loss (regularized risk minimization) and alternative multi-view learning approaches on four practical benchmarks.

Strengths: I'm broadly convinced by the results of the paper: in cases where the proximal mapping doesn't have a closed form, ProxNet does more "optimization work" on, and gets more "optimization value" out of, each example or minibatch. In cases where there is a closed form, or the inner optimization is cheap, ProxNet performs better without adding additional training cost. It's widely applicable, and the kind of idea that seems obvious in retrospect.

Weaknesses: I would have liked to see more discussion and analysis of the role of annealing lambda over training, and how the 1/t annealing schedule was chosen. Would it (for instance) be helpful to anneal the RRM term in over time as the ProxNet term is annealed out? The discussion of ProxNet equivalents to common regularizers and neural network primitives seems a little hollow: _not_ being able to construct an optimization problem whose solution is a particular given function would be the surprising case, as far as I can tell. I would have appreciated seeing multi-view learning results on larger and more recent benchmarks (e.g. semi-supervised ImageNet comparing with SimCLR), as contrastive learning seems to be picking up pace in various application areas.

Correctness: The empirical results cover a wide array of problems and include solid baseline methods and good experimentation practices (e.g. reporting mean/std). The results could benefit from more clarity about optimizers used and hyperparameter tuning (e.g., how aggressively was the lambda schedule tuned, given that it accounts for two new degrees of freedom).

Clarity: The paper is a little opaque to readers (like me) without a learning theory background, but it rewards deliberate reading as all the conceptual building blocks are included. It's also forced to be relatively concise in reporting experiment details and results, as there are four different benchmarks and three pages to fit them in. But overall the paper does a good job summing up what's essentially an entire small-scale research program with many connections to different areas of ML theory and practice.

Relation to Prior Work: The comparison to OptNet in the main text is quite limited, but there's a lot more detail in the appendix. Both OptNet and ProxNet are highly general abstractions, and ProxNet can be seen as a special case of OptNet, but the authors are clearly introducing a novel category of optimizing layer with a novel set of applications, and they discuss this fairly. Similarly, the appendix also includes theoretical treatment of the relationship with RRM (which I did not review in detail), as well as a reframing of ProxNet in meta-learning terms (again, it's a distinctly-novel special case).

Reproducibility: Yes

Additional Feedback:


Review 4

Summary and Contributions: This work proposes the use of proximal mapping to introduce certain data-dependent regularizers on neural network activations. The authors introduce two different regularization methods based on this idea. The first is a regularization on outputs of a recurrent network (LSTM) to encourage robustness to perturbations in the input. This regularizer has a closed form solution, though second order derivatives are required. The second regularization method introduced controls correlation between activations of hidden layers on two different data sets, similar to deep CCA (DCCA). The proposed method improves over DCCA by allowing the CCA objective to be jointly optimized with a final classification layers on top of the correlated representation, and is shown to be more effective than including a correlation objective as part of the overall loss. This multiview regularizer does not have a closed form solution and requires an inner optimization involving L-BFGS and SVD.

Strengths: Good results across several different tasks/datasets for each model, compared to reasonable baselines. I think these regularizers could apply to a fairly large number of problems (though not completely general like dropout etc.)

Weaknesses: Implementation appears to be quite involved, which could limit more widespread use.

Correctness: To my knowledge, everything seems reasonable.

Clarity: Reasonably clear.

Relation to Prior Work: Relation to prior work like virtual adversarial learning and deep CCA is clearly discussed. I'm not aware of any missing citations.

Reproducibility: Yes

Additional Feedback: I don't see any aspect of the robustness method for RNNs that actually relies on the recurrent structure. Am I missing something, or could it be placed on any type of neural network hidden layer?

[Author Response · NeurIPS 2020]

We thank the reviewers for their valuable comments.

**Reviewer #1:**

**Q:** Optimization of objective (22) with the proximal map (21) is not discussed in detail. MATLAB code is not available.

**A:** Thank you for bringing it to our attention. The MATLAB code has been uploaded to the GitHub repository. It
simply invokes fminunc to optimize (22), without providing a gradient subroutine. This leaves fminunc to choose its
own solver which typically utilizes its own finite difference routine. The result looks good and efficient for this dataset.

**Q:** the paper does not explore more straightforward applications to per-layer regularization in deep neural networks.
Mentioning this seemingly fundamental application, the authors do not study it just saying that it is straightforward to
implement. This makes the reader wonder whether this method can actually show promising results in this setting.

**A:** ProxNet for multi-view learning used proximal mapping for only one layer, and so did Figure 1. In contrast,
ProxLSTM in Section 4 embedded a proximal mapping into *each* step/layer of an LSTM. It will be very interesting to
study the use of multiple proximal mappings at different layers for diverse purposes, e.g., to enforce equivariance in
each layer of feature extractor by using the violation as the regularier $R$, disentanglement at a certain layer, and fairness
in prediction. This requires more space than a NeurIPS paper, and we are exploring some of these combinations.

**Reviewer #2:**

Thank you for pointing out Meinhardt et al. (2017). It is relevant and we will cite it in the future versions.

**Q:** Proximal operators may hurt the performance if they are not instantiated correctly for the problem and model that are
being considered. I do not have much intuition on how the ProxNet improvements upon other regularization approaches.

**A:** ProxNet is a means of enforcing deep regularization, while the regularizer $R$ itself is to be designed by the
practitioners for the specific application, e.g., CCA for multi-view learning. So ProxNet by itself does not introduce any
new regularizer. Implementing it is straightforward as shown in Eq (2) for any generic $R$, though some parameters
need to be chosen (see the next question). The purpose of the paper is to show that for regularizers defined in terms of
hidden layer outputs, it is more effective to regularize **in-place** through a proximal mapping at that layer, compared with
adding the regularizer to the overall objective and relying on backpropagation for optimization. By "more effective",
we have compared by using the test performance instead of the training objective value, because unlike comparing two
different nonconvex solvers, ProxNet results in a different objective than regularized risk minimization. A rigorous
analysis beyond the intuition of in-place regularization (to quote Reviewer 3: "the kind of idea that seems obvious in
retrospect") will be interesting in the deep context, and we plan to investigate it in the future.

**Q:** Does care need to be taken to properly setting the $\epsilon$ term to ensure the derivative approximation is well-behaved?
Will the finite-difference derivative approximation in L178 causes instability during learning?

**A:** [16] provided several heuristics for setting $\epsilon$. Our experiment set $\epsilon = \delta(1 + ||(X, Y)||_\infty)||(\frac{\partial L}{\partial P}, \frac{\partial L}{\partial Q})||_\infty^{-1}$ so as to be
adaptive to the magnitude of the gradient, and $\delta$ is a small constant whose value was chosen by checking a few gradients
at the beginning of training. To estimate the "true" gradient needed for the check, we used a small $\epsilon$ and solved the
proximal mapping in Eq (10) to high accuracy. With this heuristic, the approximate gradient did not cause instability in
training. It is also noteworthy that the stochastic gradient computed from a mini-batch introduces noise in the first place.

**Reviewer #3:**

**Q:** Would it be helpful to anneal the RRM term over time as the ProxNet term is annealed out?

**A:** We conducted some additional experiments to show
how an annealed $\lambda$ influences the performance of RRM.
We not only tried the annealing schedule used for Prox-
Net, but also other schedules. The results for multi-view
learning on the Sketchy dataset are shown in the right
table, where, overall, annealing has mixed influence on
the vanilla RRM, but it remains inferior to ProxNet.

| #class | 20 | 50 | 100 | 125 |
|---|---|---|---|---|
| RRM new | $17.4 \pm 0.8$ | $22.5 \pm 0.5$ | $24.3 \pm 0.9$ | $26.1 \pm 0.7$ |
| RRM old | $15.2 \pm 0.6$ | $20.1 \pm 0.4$ | $26.8 \pm 0.5$ | $28.1 \pm 0.4$ |
| ProxNet | $\mathbf{13.7} \pm 0.3$ | $\mathbf{17.9} \pm 0.5$ | $\mathbf{20.2} \pm 0.3$ | $\mathbf{22.0} \pm 0.4$ |

**Reviewer #4:**

**Q:** I don't see any aspect of the robustness method for RNNs that actually relies on the recurrent structure. Am I
missing something, or could it be placed on any type of neural network hidden layer?

**A:** The ProxLSTM in Section 4 made RNN robust by applying a proximal mapping at each layer/step for invariantization.
This is innately synergistic with the recurrent structure. In other words, we achieved robustness not because of using the
recurrent structure itself, but by properly invariantizing each step via embedding a proximal mapping. So this technique
can be generically deployed in any type of neural network.

[Meta-Review · NeurIPS 2020]

This paper discusses using proximal mappings as a regularization technique. It concretely proposes and experimentally evaluates two such regularizers: One that encourages an LSTM to be robust to changes in its inputs, and one that regularizes embeddings to be close to each other in multi-view learning. The reviewers all agreed that the method was clearly presented, well-motivated, and an important contribution. The consensus is therefore to accept.